# Revolutionising the 4D BIM Process to Support Scheduling Requirements in Modular Construction

Mohammad Mayouf [1,2,*], Jamie Jones [3], Faris Elghaish [4], Hassan Emam [2], E. M. A. C. Ekanayake [1] and Ilnaz Ashayeri [1]

1   Faculty of Computing Engineering and the Built Environment (CEBE), College of the Built Environment, Birmingham City University, Birmingham B5 5JU, UK; ilnaz.ashayeri@bcu.ac.uk (I.A.)
2   Zigurat Global Institute of Technology, 08018 Barcelona, Spain; hassan.emam@hotmail.com
3   Willmott Dixon Interiors, Birmingham B4 6GH, UK
4   School of Natural and Built Environment, Queen University Belfast, Belfast BT7 1NN, UK; f.elghaish@qub.ac.uk
*   Correspondence: mohammad.mayouf@bcu.ac.uk

**Abstract:** Given the heightened importance of revolutionising 4D BIM-based construction scheduling in modular construction, it has become vital to explore how 4D-BIM could be integrated with the lean concept. Therefore, this research aims to develop a lean-integrated process model to revolutionise the 4D BIM-based construction scheduling in modular construction projects. A case study approach was used to obtain the data. The data was obtained using semi-structured interviews with construction scheduling professionals, site observations, and extracts from the BIM model used within the selected case in the UK. Findings showed that conventional (component/object based) 4D BIM supersedes conventional scheduling methods in terms of foreseeing potential implications during design and construction. The findings also showed that lean-integrated 4D BIM in modular construction have different considerations when compared with component/object-based scheduling. A lean-integrated 4D BIM process model was developed from the analysis and it was validated using an interactive workshop with eight participants from two UK construction companies and two modular construction manufacturers. The developed process model identified a number of considerations for 4D BIM in modular projects including constructability, operations, health and safety risks and time. This study suggested the further potential of 4D BIM in scheduling for modular construction projects.

**Keywords:** 4D BIM; lean; scheduling; modular

## 1. Introduction

In construction projects, scheduling construction activities has continually been an interest for many researchers and practitioners given its significance towards achieving project objectives. In the realm of construction scheduling, various methodologies, notably the Critical Path Method (CPM) and the Project Evaluation Review Technique (PERT), are employed to effectively organise project activities and communicate them across stakeholders [1]. However, the complex nature of construction projects requires the need to communicate project information more effectively and efficiently [2]. Hence, Building Information Modelling (BIM) has emerged as an integrated and technologically supported process that can typically be deemed as an effective mechanism to foster a collaborative approach to handle interdisciplinary information efficiently [3,4]. As one of the BIM technologies, 4D BIM was introduced as an effective mechanism that integrates 3D digital construction models with time or schedule-related information, providing an efficient tool for project management and planning in the construction industry. This technology enables stakeholders to visualise the construction process over time, enhancing decision-making and communication throughout the project lifecycle. In addition, and over the years, 4D

BIM applications have been extended to cover aspects beyond scheduling, including logistics management, space management and even site development [5,6]. Although 4D BIM provides such extensive technological benefits to the construction industry, there is no known study on how 4D BIM can be revolutionised to support scheduling and planning for construction projects by eradicating non-value-adding activities from the construction schedules [7], and more importantly, to improve the decision-making process [8].

The lean construction concept has been widely used to acknowledge the ability to learn and develop continuous improvement by mitigating non-value-adding activities: construction wastes throughout all project phases and processes [9–12]. In the context of 4D BIM, the majority of studies have focused on improving workflows, overcoming scheduling issues and automating the 4D BIM process. Whilst this provides value for the utilisation and implementation of 4D BIM, it does not take into account the new thinking mechanisms that sit within the context of lean construction such as modular construction. The modular construction process is conducted within a controlled environment, in which drawings, design information, and interdisciplinary information such as production timeframes and product interface interconnections can all be integrated [13,14]. This places more emphasis on the lifecycle of a modular unit, which starts with manufacturing, followed by logistics and finally assembly. As such, it is important to utilise BIM in a way that supports recognising the associated benefits of the effective planning and delivery of modular units [10]. As a lean concept, the proper integration of the 4D BIM process requires a more careful and adapted approach to suit the requirements of modular construction projects. 4D BIM would enable a robust approach to overcome unnecessary activities in project planning while shedding the light complexities that are associated with modular construction [10,11,13,14]. The proper adaptation of the lean concept with 4D BIM capabilities in modular construction could enable the modularisation process to be as efficient as possible—from the manufacturing stage to on-site assembly [7,15]. Given the importance of filling this vital research gap, this study aims to develop a lean-integrated process model to revolutionise 4D BIM-based construction scheduling in modular construction projects.

## 2. Literature Review

### 2.1. 4D BIM: The Concept and Application

The term '4D BIM' simply refers to linking a schedule of activities to a 3D model to produce a dynamic visual representation of construction activities [16]. A 4D model requires a 3D geometric model with building components, a construction schedule/programme and a 4D environment (e.g., a software application or an appropriate interface) that enables linking the 3D model with the construction schedule to create the 4D simulation [17]. Over the years, 4D BIM has benefited the construction industry in terms of enhancing the coordination and visualisation of the schedule for both existing and new projects [18], which enables professionals to communicate through a digital model [19], creates interactive simulations for construction progress, supports managing different aspects including logistics and site facilities, and provides a more proactive mechanism to communicate with the client.

More precisely, as a significant element of 4D BIM, the visualisation functionality was used to detect conflicts between workspaces, analyse progress, and provide space management [20–22]. In fact, some research efforts showed that 4D BIM can support more informed decision making, which can be in terms of site layout [20], logistics and space management and can even extend to the analysis of constructability [6]. It can be stated that 4D BIM acts as an effective mechanism to support project goals, provide more transparency and for more informed coordination of site activities. Although some research attempts are continuing to demonstrate the potential of 4D BIM, most of the developed tools are context specific [7,16,23,24]. Recent research showed that there is a need to advance the use of 4D BIM so that its value can be maximised in construction projects [7,24,25]. The value of 4D BIM can be maximised in the planning and scheduling process by avoiding non-value-adding activities, so-called wastes from construction

schedules. However, there is an absence of literature on how to eradicate non-value-adding activities from construction schedules and optimise the 4D BIM-based scheduling and planning process. In this regard, lean construction concepts could play a vital role given their strong focus on waste elimination and achieving perfection, described as follows.

### 2.2. Lean Construction: Concept and Application

In construction, the use of "lean" is a thought process often targeted but rarely fully achieved. It is a thinking methodology that can be applied to construction and design principles to minimise material waste, and generate maximum value output, whilst enhancing the efficiency of activities on site [25]. Broadly, lean is underlined by many principles, which is often abstracted through many processes [11,26]. There are 11 core principles within lean: reduce non-value-adding activities, increase output through systematic consideration of consumer requirements, reduce variability, reduce cycle times, simplify by minimising the number of steps, parts and linkages, increase output flexibility, increase process transparency, focus control on the complex process, build continuous improvement in the process, balance flow improvement with conversion improvement and setting a benchmark [12]. The literature also abstracted lean principles into six processes: defining the value stream, eliminating waste, monitoring work processes, pull planning and scheduling, identifying value from a customer point of view and continually improving processes. Perhaps the core principles of reducing, eliminating and improving can be used as the underlying principles for embedding lean within construction projects.

In construction projects, applying lean is not restricted to a particular stage or phase, but it can be a running thread from pre-design, procurement, construction processes and even operation. There are many techniques that can be applied during design (e.g., cross-functional teams, set-based design and design for buildability), procurement (e.g., supplier training, work packaging), material handling (e.g., just-in-time and elimination of packaging) and operation (e.g., multiskilled craftworkers) [27]. Although these techniques were outlined over two decades, a recent review revealed that many of nowadays' obstacles to implementing lean in construction projects can be traced to the lack of understanding of basic principles [28]. As such, it is essential to have a better idea of how to implement lean effectively within the construction project processes. In addition, lean is widely applied and established in the manufacturing industry given its significant merits in production performance improvement. Therefore, it is well applicable to the modular construction processes as the modules are manufactured at prefabrication factories and follow a similar manufacturing workflow. Further, there are several advantages of deploying lean-integrated 4D BIM in modular construction projects as discussed below.

It can be stated that 'lean' principles in construction represent a transformative approach, yet achieving its full potential remains a challenge. Lean methodology, with its emphasis on minimising waste and maximising value, fundamentally alters on-site efficiency. However, its application extends beyond theoretical principles; it requires a nuanced understanding of the construction environment. The core principles of lean, ranging from reducing non-value activities to fostering continuous improvement, are not just theoretical constructs but practical tools for enhancing construction processes. While principles such as reducing cycle times and increasing process transparency are universally applicable, their impact is contingent upon the project's context and scale. A critical gap often lies in the translation of these principles into actionable strategies during different construction phases, from design to operation. Moreover, the evolving landscape of construction, marked by a heightened focus on sustainability and digital integration, demands an adaptive approach to lean methodologies. This aligns with recent literature which indicates a gap in understanding and implementing lean, pointing to a need for more tailored, context-specific applications in construction projects.

### 2.3. Lean Integrated with 4D BIM for Modular Construction

Modular construction falls within off-site manufacturing (OSM), which is economically beneficial, provide efficient speed for installation on site, and improves quality and quality control. According to Lawson et al. (2014), off-site manufacturing support many module units, and in some cases produces a completion of up to 95%, before being shipped and installed on site, to add the final finishes of 5% [29]. More importantly, it creates an optimisation culture of efficiency, technology integration, speed, labour, resource and cost [30]. The practical implications of off-site manufacturing are that it potentially supports minimising the programme and scheduling time by reducing the installation of many components on site. Bearing this in mind, there are still risk factors associated, which include damage during transit and the potential for cost-saving materials to be used, which may not be as robust as other materials. Nevertheless, recent research showed that as a result of off-site manufacturing methods such as modular construction, the level of productivity and process efficiency in construction projects have increased [31,32]. To create a meaningful synthesis, and for this research, Figure 1 shows synthesis from the literature connecting lean concepts to modular construction components where impacts on cost and activity planning would be.

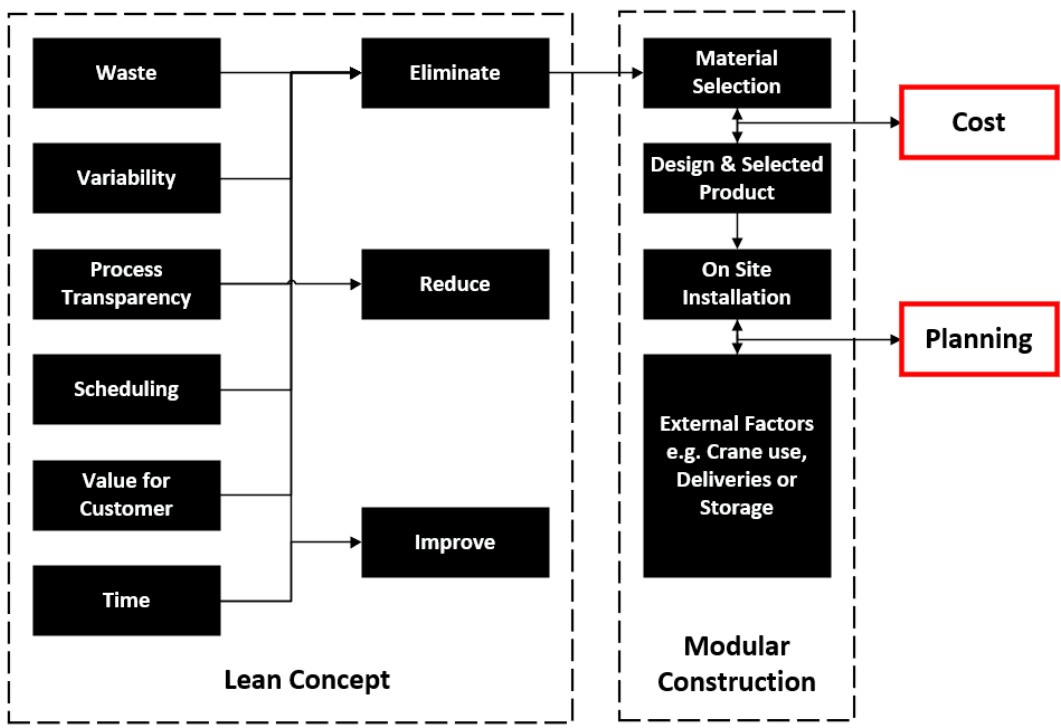

**Figure 1.** Synthesis of lean concepts, modular construction illustrating where implications on cost and activity planning would be.

In the context of BIM, there is a growing interest in integrating the use of 4D in lean construction. For instance, a study by [33] proposed a BIM-based Last Planner System (LPS), which enable project managers to visualise processes and operations generated in the schedule with their corresponding BIM objects. Another study by [34] highlighted that 4D BIM for modular units should aim towards improved management of tasks on site. The study highlighted the necessity to align 4D capabilities to support critical operations such as crane lifting, safety monitoring and motion of heavy machinery. A recent study by [35] suggested the need to identify 4D BIM objectives in a project in order to structure appropriate workflows that responds to project requirements. The above studies, whilst illustrating significant advancements in 4D BIM, portray that defining 4D objectives in a

project is paramount, and this necessitates the need to insightfully explore it within the context of modular construction.

Modular construction, as a subset of off-site manufacturing (OSM), presents unique opportunities for the integration of lean principles, especially when combined with 4D BIM technologies. The literature shows that an appropriate optimisation in modular construction lies in the intersection of lean practices with 4D BIM. This integration not only streamlines the manufacturing process but also enhances on-site installation efficiency. The real-time visualisation and planning capabilities of 4D BIM, when aligned with lean's waste reduction and process optimisation strategies, could potentially address common challenges in modular construction, such as logistical complexities and on-site assembly inefficiencies. However, this integration is not without its challenges; it necessitates a nuanced understanding of both lean principles and 4D BIM capabilities. While current research highlights the productivity benefits of such integration, there is a need to explore how this synergy can be tailored to different scales and types of construction projects. The potential for lean-integrated 4D BIM in modular construction is immense, yet its application must be contextually grounded and adaptive to the evolving technological landscape in construction.

### 3. Methodology

This study adopts a qualitative research approach, employing a case study design to investigate the development of a lean-integrated process model aimed at transforming 4D Building Information Modelling (BIM)-based construction scheduling in modular construction projects. Recognising the multifaceted and dynamic nature of this field, characterised by intricate interactions, processes, and human factors, the qualitative approach is particularly suited for exploring and understanding complex phenomena where contextual nuances are essential. The rationale for selecting a case study design lies in its ability to facilitate theoretical generalisation, aligning with the primary aim of developing a conceptual framework with broader applicability. This approach enables the contextualisation of diverse data sources, yielding a comprehensive narrative that enriches our understanding of the research inquiry.

Furthermore, the case study design enables the investigation within the real-world setting of an ongoing University Library Building project in the UK, providing vital contextualisation for a deep and holistic understanding of lean principles integration into construction scheduling using BIM, specifically within the modular construction context. This chosen case involves the construction of a University Library Building in the UK, noted for its pioneering use of lean-integrated 4D BIM in the modular construction domain.

Data acquisition was achieved through a triangulated approach encompassing semi-structured interviews, on-site observations, and analysis of BIM models. Triangulation strengthens the validity and reliability of findings by corroborating information from multiple sources, thus reducing potential bias or misinterpretation. For the semi-structured interviews, interviewees were selected based on their direct involvement and expertise in the ongoing University Library Building project in the UK, ensuring a deep and informed perspective on the subject matter. The interviews were structured around a guide, yet flexible enough to enable in-depth exploration of topics as they arose naturally. The focus topics included experiences with project scheduling through BIM, the integration of lean principles into scheduling processes, and the application of 4D BIM in modular construction. Interviews were audio-recorded, with consent, and later transcribed verbatim to ensure accuracy in data analysis. In total, four construction planners were identified (those who were involved in the delivery of the University Project) who shared their experiences with project scheduling through BIM, the infusion of lean principles, and the application of 4D BIM in modular construction. On-site observations documented photographs, field notes, and issues encountered during modular unit assembly, contextualising the research within the actual construction environment. Analysis of the project's BIM models provided a visual representation of specific challenges, enhancing spatially contextualised issue discernment.

It is important to note that the use of BIM extracts was restricted to 1–2 examples due to intellectual property constraints. The triangulation of data collection methods, combining semi-structured interviews, field observations, and analysis of BIM models, enhances the validity and reliability of the findings. This multimethod approach mitigates potential biases and offers a comprehensive view of the case study.

A rigorous thematic analysis methodology was employed to scrutinise the data. This analysis involved a line-by-line coding approach, identifying recurrent themes and patterns. Categories from site observations and BIM model extracts were aligned with emergent thematic structures. After analysis, a comprehensive lean-integrated 4D BIM process model was formulated, encapsulating case study insights. Validation of the process model was conducted through an online interactive workshop with industry experts, ensuring the model's practical utility in enhancing construction schedules. Participants included construction planners, BIM managers, and modular unit manufacturers, each contributing unique perspectives, thereby enriching the validation process.

In summary, this study's qualitative approach, through its case study design and data triangulation, enhances the depth and breadth of findings, aligning with the over-arching research goals. The comprehensive approach addresses the complexity inherent in the research subject, ensuring robustness and relevance in the findings, while acknowledging the subjective nature of qualitative analysis. The inclusion of diverse perspectives and the systematic approach to data analysis and validation further substantiate this study's findings. Table A1 contains detailed responses from construction planners, BIM managers, and modular unit manufacturers, labelled as CP, BM, and MUM, respectively, providing transparency and further contextualisation. Figure 2 presents the comprehensive research design followed in this study.

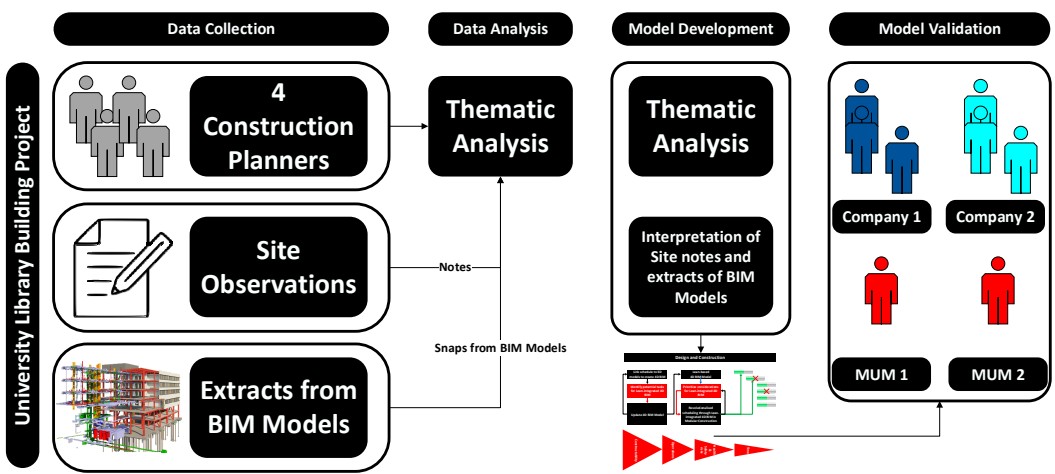

**Figure 2.** Research design employed in this research.

## 4. Results and Analysis

The data gathered using semi-structured interviews was analysed using line-by-line coding (as informed above), and a total of three themes were identified, and explicated as below. The three themes are: 1. Traditional vs. 4D BIM scheduling 2. Lean-integrated 4D BIM and 3. 4D BIM in modular construction.

### 4.1. Traditional vs. 4D BIM Scheduling

This theme aims to shed the light on optimum processes that support the construction scheduling team in a project. As part of the initial enquiry within this theme, it was looked into as a sensitive thread that makes or breaks a schedule in construction, hence participants were asked to indicate whether traditional or 4D BIM would provide a better means to track planned vs. actual progress. Participants have demonstrated varied perceptions of planning and scheduling on construction projects. Participant 4 shared their views of the

accuracy of planning and scheduling; "*No I don't feel planning and schedules are accurate, with the lack of taking a programme and then turn that into a plan which is achieved a major issue. Currently, I see the same mistakes being made across UK construction projects, with little evidence of learning*". This is instigating that "we" as an industry need to develop and learn new solutions to planning and scheduling to help the understanding of how best to change plans. As for 4D BIM, Participant 1 stated "*4D should be seen as better planning, in any case*". However, the participant was able to distinguish that you could not move from traditional planning such as the Critical Path Method (CPM) "*you wouldn't traditionally (CPM) plan a project and then move to 4D BIM, the 4D process would need to be undertaken from the start*". This not only highlights that 4D BIM has the perception of being a 'better' planning and scheduling technique but also it must become culture driven from the start of a project; otherwise the combination of planning techniques can create a negative flow. Participant 4 further confirmed that droplines on the CPM of planning do not always give full clarity to all involved; "Being able to visually show progress at dates on a programme as opposed to just a drop line, would be a huge step forward and allow for the site teams to understand better what we are trying to achieve".

As the project used modular construction, Participant 3 indicated "*time and cost perspectives with risks being moved outside of the site boundary*". The over-arching result of good planning and scheduling is to not over-complicate activity schedules; it is the ability to secure a target date rather than shortening the programme, as suggested by Participant 1; 'We tend to suggest merely securing the end date is as good as shortening a programme, the team are exposed to the detailed plan of work visually, allowing them to communicate concerns or ideas that might be missed in large complex Gantt charts. However, Participant 3 indicated a key BIM aspect which leads to the ability to produce shortened programmes through the use of Clash Detection, in which design elements which would clash upon installation interfaces on site, causing defects and delays within rectification periods can be developed out of the scope of works, 'the most basic one used is clash detection, being able to recognise and resolve those issues before occurring on-site saves a load of time for us'. However, on realistic value from 4D BIM, Participant 2 indicated "*there is a link before ironing out the design before getting on-site, that process of clash detection is still a 3D activity which will influence the programme majorly if not recognised, therefore early identification means that it helps tie in with the 4D aspects, it influences programme lengths, material placement, installation and events. If you can plan material layoffs, that will be brilliant, it does shorten programme lengths*". On a similar line, Participant 4 indicated "*4D process alone cannot make vast improvements, and it is with a cultural change and embracing of technology, No system in the history of the industry has single-handedly improved anything, it's the cultural change of embracing the technology and using it differently that delivers benefit. It's just a pretty tool to programme with, what you do with it will determine how good it will be*".

It is therefore clear that professionals' views of the traditional Critical Path Method of planning and scheduling do not take into account or reflect the implications of design issues, material laydown, labour installation, or at what point certain aspects must be considered at the site level. The Critical Path Method (CPM) creates a programme which could be argued does not include enough external considerations to produce an accurate programme, as such, it is a recommendation of the professionals' to use 4D planning, as it can consider fundamental considerations as well as external factors; to make informed decisions at how best a schedule can be put together and a programme produced.

In summary, the responses suggest a growing recognition of 4D BIM's potential to enhance planning accuracy and clarity. However, a shift to 4D BIM requires a cultural change from traditional CPM, underlining the need for early adoption in the project planning stages.

### 4.2. Lean-Integrated 4D BIM

Concerning lean construction and processes, most participants indicated a positive attitude towards lean construction. For instance, Participant 1 stated, "*If I'm honest I'm not*

*overly confident or know what lean construction is*". Participant 2 added "*The idea is very good, but it's difficult applying aspects from a factory environment to a project e.g., at New Street Station in Birmingham, they managed to shorten a lot of projects, but introducing ideas such as working day and night. The question is asked can you apply factory aspects to projects, such as; towards day by day programmes and subcontractors (small tweaks)*". On the other hand, Participant 4 pointed out that "*Lean inclusion in the project is often misunderstood and not used correctly across the industry*". In fact, Participant 3 mentioned that "*Lean Construction for me, means looking at efficiency across a project, in the apparent cases of material wastage, but also excessive resourcing of tasks etc.*".

One of the questions asked during the interview focused on the participants' knowledge of the seven wastes (transportation, inventory, motion, waiting, over-processing, overproduction and defects) using observations on site. The results showed limited awareness of waste management by most participants, and their knowledge was quite generic. Although the responses towards lean construction were quite generic, they reflected a limited understanding of the wider image that lean encompasses. This was further evidenced when mentioning the participants' familiarity with the 5 "S"(sort, straighten, shine, standardise and sustain) strategy for lean incorporation on site. Reflecting on this, Participant 3 indicated "*I am not familiar with the 5's. Looking over the list, I think this points to better planning practices, which goes hand in hand with 4D modelling and the benefits it brings*", whilst Participants 1 commented "*While we acknowledge the importance of integrating lean into our projects, this is considerably complex*". On integrating lean in 4D BIM, Participant 2 stated "*We use 4D BIM to support our scheduling process, but our focus is often on delivery and avoiding delays, and it is difficult to recognise lean principles of lean as part of the 4D BIM process*". What was also noticeable throughout the research, was the participants' understanding of the link being made between lean and 5 "S"; Participant 1 stated "*The lean principles apply to the 5 "S", e.g., Straighten—Defining the value stream, (lean principle). You can look at the lean principles and apply those to the 5's very easily, I can certainly identify the links without knowing too much about lean construction*". This confirms that the link between the two is simplistic; therefore, inheritance and delegation of the 5 "S" and lean theory to lean incorporation and practice on a project should be an aspect that is easy to achieve, and thus impact planning and scheduling. Participant 3 suggested, "*We very much apply these already, perhaps not in name, but by encouraging teams to adopt better processes and approach to their plan of work, which can potentially improve the way we utilise 4D BIM*", suggesting the principles of the 5 "S" are already being incorporated from a 4D planning point of view, which shows that applying lean-based techniques such as modular construction can indeed add value.

In summary, while lean construction is acknowledged as a valuable concept, there is a gap in its comprehensive understanding and implementation within the industry. This indicates an opportunity for more focused education and training in lean construction principles. More importantly, in the context of 4D BIM, based on the analysis, it is recognised that there is a lack of incorporating lean principles of lean within the 4D BIM process. The participants also added that whilst lean principles are acknowledged within processes and workflows, incorporating these principles as part of 4D planning remain complex and yet to be tested.

*4.3. 4D BIM in Modular Construction*

The final theme focuses on the potential of lean processes, which is modular construction in this research, to support improved activity scheduling and planning, and how this can be facilitated more effectively using 4D BIM. To gain a logical narrative, participants were asked about the potential of lean processes towards scheduling and planning, and Participant 3 indicated "*I think we'd be able to eliminate 95% of the wastes, depending on Gantt chart detail and density, as the typical highest granularity is down to a daily. By visualising the working flow, material deliveries, access, resourcing, these issues would be flagged and identified in a digital rehearsal ahead of arriving at the site*". When linking lean construction with 4D BIM, and within the context of modular construction, Participant 1 stated "*I think with*

*the 4D yes because you can foresee a lot of that stuff, especially with the inventory and material layoffs and definitely with the crane usage, not with the defects as that falls to human error and that falls back to unforeseen circumstances*". Participant 4 added "*With modular, modular isn't always the best route for every project, with the project we are on now it wouldn't help us at all, but you could definitely use the 4D to eliminate many of the 7 Wastes, but it always goes back to that question of how much can you foresee and unforeseen? I think definitely, for waiting and inventory but not for overproduction and defect, there needs to be that balance, it can only do so much*". It can therefore be interpreted that with the lean thinking and incorporation of 4D and modular construction, many of the wastes can be removed from an on-site perspective, tested digitally and proving ideas and theories and aspects of the construction through a digital realm of 3D/4D BIM before starting on-site installation. In demonstrating some of the practicalities that 4D BIM can support with relation to modular construction, Participant 2 pointed out "*With BIM and the digital platform with real-time data allow for us to understand progress and react in real-time, we need to move away from measuring in weeks & days lost and start to move toward hours. This will be easier through the use of BIM and will help us to align processes accordingly*".

To gain richer insight into the potential of 4D BIM with modular construction, snapshots were obtained from the BIM model used for the project. With the use of modular construction, it was identified, for instance, that associated risks with the installation of Façade Structural Framing (see Figure 3) can vary when compared with traditional component-based modelling and installation of façade structures. For instance, in component-based modelling, the emphasis of 4D BIM will be upon time, resource allocation, sufficiency of the level of detail in a BIM model, and how it interconnects with the respective activity in the schedule. On the contrary, when using modular units for façade structures, the emphasis within 4D BIM will be more upon constructability, safety procedures for assembly and level of accuracy of the unit and how it is positioned in the BIM model. This reflects the importance of recognising the difference in emphasis between component-based 4D BIM scheduling and modular unit-based 4D BIM scheduling.

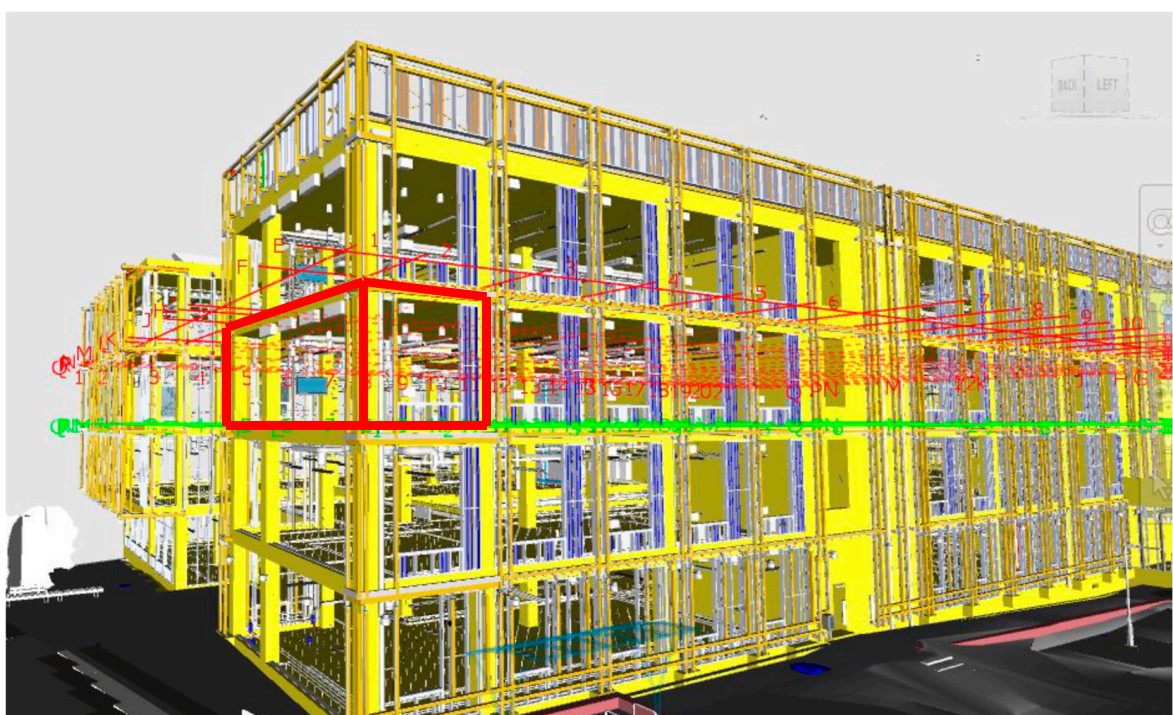

**Figure 3.** Extracts from the BIM model showing a modular unit of a Façade Structural System (inside the red box) located at upper levels, which can be associated with many risks such as constructability and safety procedures.

Another example, which represents a more critical aspect of many construction projects is the installation of plants within the site. The criticality of these elements can be envisaged through the connections (see Figure 4) they require from the top of the building and how they connect to other services in the building. In many instances, and as a result of any inaccuracies that occur on site, these plant units would change and may require design alteration, which can be an additional generator for waste, and often result in delays.

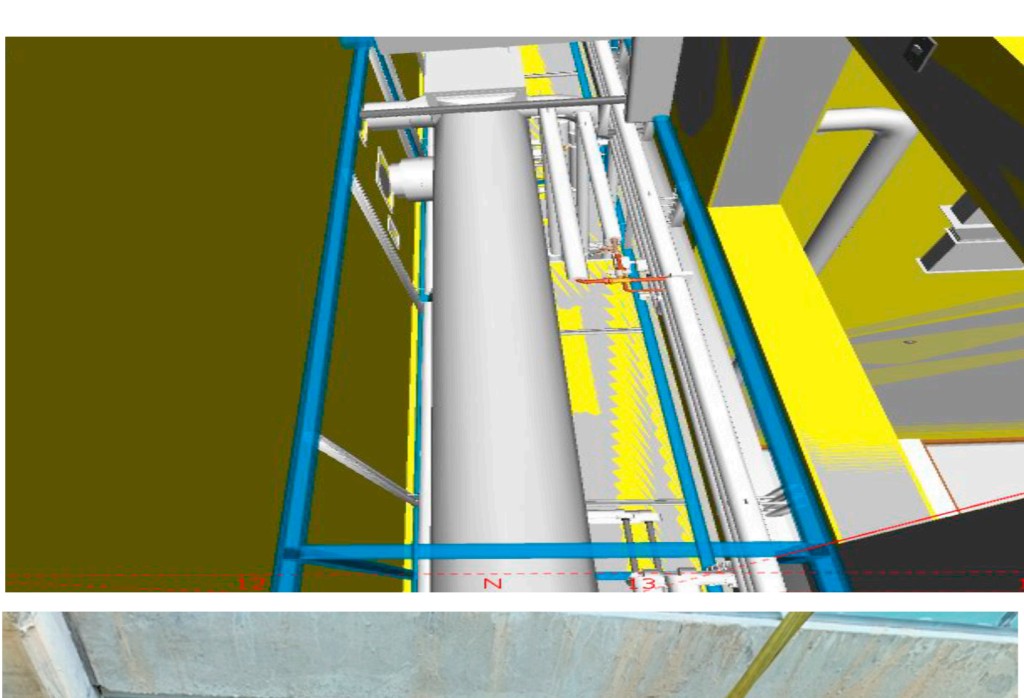

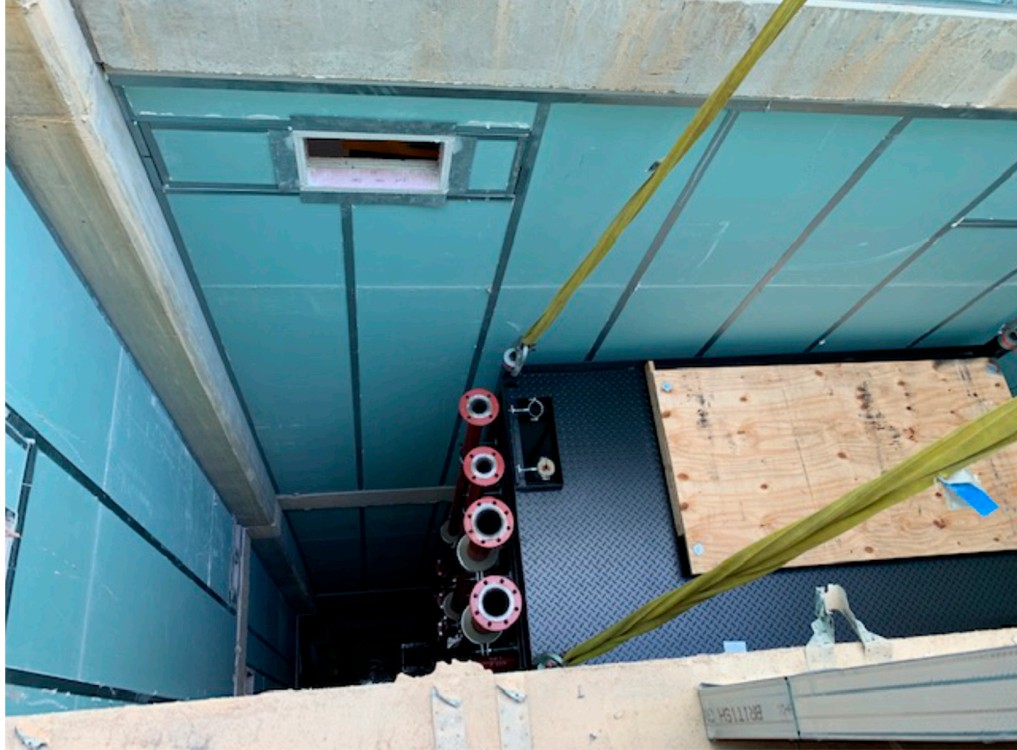

**Figure 4.** Mechanical Riser Module after installation on site (**bottom** image) and how they are represented in a BIM model (**top** image).

Based on the instances provided, it can be stated that using 4D BIM to help achieve lean processes such as waste identification on site, through the 5 "S", can have an impact on the planned vs. actual progress identification. The optimal level would be a 1:1 ratio in terms of 1 activity planned and 1 activity achieved (within the denoted programme

timeframe); however, the more planned activities there are, the more diverse the range of actually completed activities would be if the risks or wastes cannot be reduced. Reflecting on this, Participant 3 stated " *I'd suggest that this is a retrospective way to measure if a plan is 'optimal', as it may not be tracking an optimal baseline, and once you'd see the baseline moving away from the actual, it is often more difficult to recalibrate the plan*". Participant 1, on the other hand, argued that "*4D BIM isn't necessarily at the level of detail required to be able to schedule a detailed planned vs. actual percentage, but with deploying more determined objects in the model such as modular units, there is more room to have more informed consideration for labour and machine force required to optimise the volume of works on-site*". In elaborating further, Participant 2 mentioned that "*In projects, we planners propose schedules, judge on duration and sequence of activities, but in many instances, there is a high level of subjectivity and this is because human error and misjudgement are involved*". Linked to this, one of the benefits of using modular units as part of the project is mitigating health and safety risks when installing risers (see Figure 4).

Referring back to Figure 4, and as reflected from on-site observation, the installation of Mechanical Riser Module Units encapsulates a multifaceted array of risks. Paramount among these is the lack of accountancy towards structural integrity, a consequence of the prefabricated nature of Mechanical Riser Module Units, which, if improperly manoeuvred during the lifting and installation process, can compromise their integrity. This is compounded by the susceptibility to installation errors, potentially culminating in long-term ramifications such as misaligned connections, leakage, or suboptimal operation of the encased mechanical systems. In the process of installing within the chosen site, the process of elevating and positioning these substantial units necessitated the use of cranes and analogous lifting apparatus, thereby introducing a spectrum of handling risks. However, in the visual representation in the BIM model (see the top image in Figure 4), the representation of such mechanical components is complex due to the amount of fittings and number of connections with Mechanical Risers in other modules.

Expressing the potential value of modular construction and 4D BIM, Participant 4 indicated that "*You could massively improve on construction project timelines; I do feel though that we may find the gains are lost in projects which do a bit of modular construction and lose time in other areas of the programme. We need to be clear in measuring and reporting the benefits so that we can push the process to be 80% of the programme as opposed to 20%*". Compared to the traditional Critical Path Method of planning, Participant 3 included aspects relating to "*I think more. As modular tends to dictate more structure and order to the construction sequence, whereas traditional allows for more flexibility. But by using 4D this is hopefully easier to manage, earlier*". Participant 2 concluded that "*I think we will have fewer site-based activities that sit on the critical path there are still going to be issued in the factory before coming to site in essence moving the critical path from site to the manufacturing factory*". It can therefore be stated that lean-based processes such as modular construction can provide, beyond reducing waste, potential towards mitigating many risks but, more importantly, can support more informed decisions on activity planning and scheduling on site. This can perhaps act as a solid ground to enable technologies such as 4D BIM to provide a medium to incorporate modular-related considerations for scheduling and activity planning.

In summary, based on participants' responses, it is recognised that the integration of 4D BIM with modular construction is perceived as a significant step towards more efficient and safer construction practices. However, the extent of its impact varies based on project specifics, indicating the need for adaptable approaches in different construction scenarios. In the context of modular construction, different participants elaborated on different parameters that directly impact modular units on site including constructability, operational handling, health and safety and time.

## 5. Discussion

### 5.1. 4D BIM: Beyond Scheduling Potential

From the analysis, it can be summarised that, from the participants' perspectives, the use of 4D BIM, if utilised, must be driven from the beginning of a project and should be

perceived as a culture. It is only at the point at which the use of 4D BIM becomes a culture that all information sharing, collecting and updating will be directed through the BIM process and model [16,36]. The 4D BIM process exists primarily to enable improved coordination between site teams to be able to visually see the construction process of a project. This in essence enables "coordinated project information repository", which contributes towards a better understanding of the plan, the time parameters and the timeframes involved from "calculations of durations using automated quantity extraction processes", and the sequence involved, whilst identifying the risks through model assessment and discussion, increasing collaboration and giving opportunity for those involved to share ideas or concerns [37]. In fact, and reviewing the literature, it can be realised that over the years, whilst the application and potential of 4D BIM are on the rise [7,16,18,38], the analysis from this article revealed that considerations towards 4D BIM for modular construction need to differ when compared with traditional construction. In the context of modular construction, the analysis revealed that 4D BIM enables teams to visualise the assembly and integration of prefabricated modules, aligning with lean's emphasis on efficient workflow and continuous improvement. Moreover, it facilitates better planning and coordination of the off-site fabrication of modules with on-site activities, ensuring a smoother integration of components. This is particularly beneficial in identifying potential logistical challenges and bottlenecks, which are critical in modular construction due to its reliance on timely delivery and assembly of modules.

It is therefore essential to position the method of construction as the primary driver towards the use and consideration behind 4D BIM in a project. In that respect, there are some recent studies (e.g., [23]) that began to look into modular-specific considerations in 4D BIM such as safety management for assembly [7,16,18,23]. The adoption of lean principles (e.g., eliminate, reduce and improve) through 4D BIM in modular construction also aids in streamlining supply chain management. The visualisation and scheduling capabilities of 4D BIM enable precise forecasting of material requirements and just-in-time delivery, which are key tenets of lean. This integration not only reduces storage and holding costs but also minimises the risk of material damage and loss. In essence, the fusion of 4D BIM with lean principles in modular construction transcends traditional scheduling benefits. It fosters a culture of efficiency, precision, and continuous improvement, making it an indispensable tool in modern construction methodologies. The next section elaborates further on the recommendations towards deploying a more effective and efficient utilisation of 4D BIM within modular construction-based projects.

*5.2. Lean-Integrated 4D BIM in Modular Construction*

Based on the primary data analysis, this research proposes a process model that outlines the considerations for lean-integrated 4D BIM in modular construction (see Figure 5). The analysis highlighted that 4D BIM considerations for modular construction differ to those that concern component/object-based approach (e.g., assigning a 3D model component to a construction activity). The research also elaborated on considerations for modular units within 4D BIM environments, which are often concerned with aspects such as constructability, health and safety, operations and also time. To convey meaningful outcomes from the presented research, considerations identified, based on synthesis from the literature and primary data analysis, were structured in a logical order to inform the 4D BIM process in modular-based projects. Figure 5 provides a basis to support lean-integrated 4D BIM in modular construction and lays a more informed ground for scheduling considerations. This process model can also complement many on-going efforts (e.g., [23,32,39]) that looked into 4D BIM in modular projects. The approach in this research further sheds light on object-based complexities in 4D BIM and more importantly overcomes many of the design inefficiencies [7,40].

Figure 5 shows the proposed process model that supports achieving a modular-based scheduling and planning process using 4D BIM. To develop the proposed process model, the authors synthesised findings from the primary data and relevant studies from the

literature. The three themes identified a number of considerations including complexities related to the implementation of 4D BIM and parameters that impact modular construction on site. Additionally, an in-depth literature review was conducted (using studies mentioned in the previous section) to understand existing frameworks and practices in lean construction and 4D BIM, focusing on modular construction. Based on the synthesis conducted, the process model highlights considerations for lean-integrated 4D BIM, where the considerations identified were constructability, operations, health and safety risks and time. The constructability analysis can support identifying complex modular units. This can mitigate many of the issues that can be faced when planning different operations (e.g., logistics and resources) for assembling and installing modular units on site. In a recent study by [34], it was indicated that modular units in 4D BIM should focus on complexity of the object, location, installation requirements and even movement of lifting equipment. Following the identification of operations-related concerns, potential health and safety risks can be highlighted and appropriately updated within the 4D BIM model. This will support efficient and more informed construction schedules by removing unnecessary activities or re-arranging activities so that delays can potentially be reduced. This proposed model, therefore, avoids potential duplication of activities and reduces clashes that can occur, especially during assembly on site.

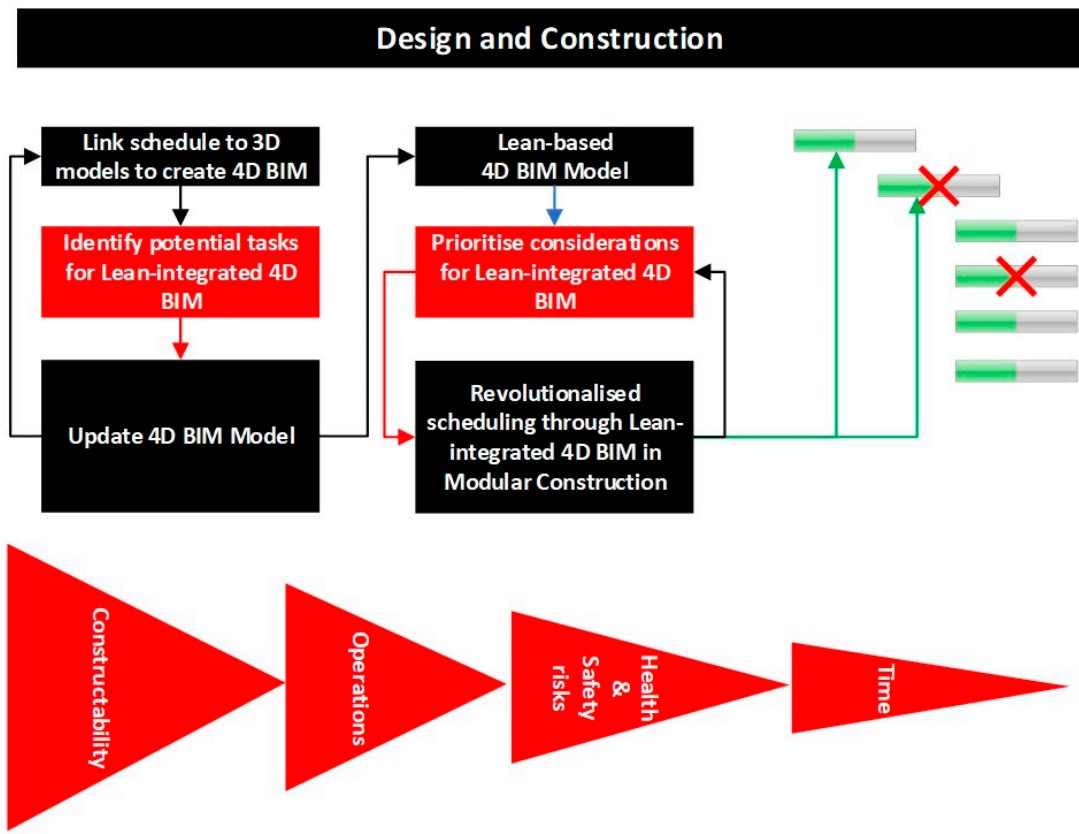

**Figure 5.** The proposed process model to revolutionise 4D BIM for modular construction.

## 6. Conclusions

To conclude, this paper successfully developed a lean-integrated process model aimed at revolutionising 4D BIM-based construction scheduling in modular construction projects. While the literature demonstrates various evolutions in 4D BIM applications, its application within modular construction remained notably underexplored. Our qualitative approach, encompassing a case study of the University Library Building, utilised semi-structured interviews, site observations, and BIM model extracts to gather rich, contextual data. Our analysis revealed distinct considerations for 4D BIM in modular construction, diverging

from traditional component- or object-based approaches. The proposed process model, specifically designed for modular units in 4D BIM, was validated through an interactive workshop with industry professionals, highlighting its practical applicability. This study not only underscores the untapped potential of 4D BIM in modular construction but also proposes a model that supports more efficient and informed scheduling, with the capability to minimise delays through the elimination or rearrangement of activities.

However, it is important to acknowledge certain limitations of this study. Firstly, the case study specificity, centred solely on the University Library Building project, may limit the generalisability of our findings across the diverse spectrum of modular construction projects, each with their unique complexities, scales, and contexts. Secondly, the qualitative nature of our research, while rich in context, lacks the empirical rigor and statistical validation inherent in quantitative methods. This may constrain the ability to quantitatively measure the benefits or efficiency improvements introduced by our proposed model. For future research, a comprehensive evaluation of this model in live project environments is planned. This will further assess the impact of integrating lean principles with 4D BIM on modular construction scheduling practices, potentially offering transformative insights for the field. Future research looks into assessing the applicability and impact of the proposed model in live modular construction projects, as this will provide real-world evidence of its effectiveness. Additionally, the scalability of the proposed model for different sizes and types of modular construction projects and its adaptability to different projects will be investigated.

**Author Contributions:** Conceptualisation, M.M., J.J., F.E. and H.E.; methodology, M.M., F.E., E.M.A.C.E. and I.A.; validation, M.M., J.J., H.E. and E.M.A.C.E.; formal analysis, M.M., H.E., E.M.A.C.E. and I.A.; data curation, M.M., J.J., F.E., H.E., E.M.A.C.E. and I.A.; writing—original draft preparation, M.M., J.J., F.E., H.E., E.M.A.C.E. and I.A; writing—review and editing, M.M., J.J., F.E., H.E., E.M.A.C.E. and I.A. All authors have read and agreed to the published version of the manuscript.

**Funding:** This research received no external funding.

**Institutional Review Board Statement:** Not applicable.

**Informed Consent Statement:** Informed consent was obtained from all subjects involved in this study.

**Data Availability Statement:** Data are contained within the article.

**Conflicts of Interest:** The authors declare no conflict of interest.

## Appendix A

**Table A1.** Responses received during the workshop by the different participants to validate the proposed process model.

| Question | Responses |
| --- | --- |
| Does the proposed process model provide an improved mechanism to tackle complexities associated with construction schedules? | "In a design and build scenario, schedules created during the tendering stage is often revised at later design stages, and this results in many issues, so with this process model, having a progressive construction schedule during the design will help to detect constructability issues" (CP 1) |
| | "BIM Models are complex, and they differ in every project, and whilst we often focus on design parameters, we often forget about constructability issues, which I agree can be significantly reduced with Modular Construction" (BM 2) |
| | "Constructability is really a huge issue, and it has major cost implications on different operations which we as construction planners have to deal with, so this process model would help to prompt the attention toward complex design elements using 4D BIM" (CP 3) |
| | "Collaboration is a key, and current use of 4D BIM models is often during later design stage, so I would agree that such process would help reducing many of the complexities within construction schedules" (CP 2) |
| | "Modular unit manufacturing often requires precise coordination to meet project deadlines. This process model would enhance our ability to plan and execute the manufacturing phase in alignment with the construction schedule, reducing various health and safety risks and on-site complexities." (MUM 1) |
| | "Efficient scheduling is crucial in modular construction, and this process model's emphasis on early detection of constructability issues would greatly benefit our manufacturing process and ensure smoother integration on-site." (MUM 2) |

**Table A1.** *Cont.*

| Question | Responses |
|---|---|
| Does the proposed process model provide more potential to revolutionise the use of 4D BIM for Modular Units? | "4D BIM Models are often used to visualise the process, and in very few instances, to detect issues, so with this process model, there is a potential to provide more focused insight into complex design elements especially for superstructure elements and services" (CP 3)<br>"We usually spend long time in mapping BIM Models with construction schedules, and with all the changes, it becomes very difficult to have as-designed model to be used for scheduling, so with this model, construction planners can advise when multiple disciplines are in a single 3D model" (BM 1)<br>"4D BIM models should be used for decision-making, so this model support progressive use of 4D BIM Models to have more robust and easy to follow mapping between 3D objects and construction activities" (CP 2)<br>"3D Modelling consumes both time and effort, but you can only visualise their complexity when using 4D BIM Models, and this becomes more complex when multiple parties are involved in the design phase, so with this process model, 4D BIM would definitely become more beneficial" (BM 2)<br>"Modular units may not be suitable for every project, but often thinking about them happen before the schedule is prepared, so I agree that 4D BIM can potentially support recognising whether Modular units are suitable, as this will reduce many of the operations and constructability issues during the construction phase" (CP 1)<br>"Manufacturing modular units involves various disciplines and components. This model's focus on mapping 3D objects to construction activities is promising, as it would enable us to align our manufacturing processes with the broader project schedule more effectively."<br>(MUM 1)<br>"The proposed process model appears to encourage a more strategic use of 4D BIM in modular unit production. It could help us make data-driven decisions, streamline our manufacturing processes, and ensure that our units are delivered on time and with high quality."<br>(MUM 2) |
| Do you think a Lean-based 4D BIM Model is beneficial for construction scheduling? | "In construction schedules, we spend long time identifying appropriate resources, requirements and additional measures, but with lean-based BIM Model, the number of requirements for Modular-based activities will reduce and the use of 4D BIM will be more beneficial" (CP 4)<br>"4D BIM models are very good for visualisation, but they do not really help to schedule activities, so I think with the use of Modular units, 4D can become more useful and easier to map with construction schedules (CP 2)<br>"Construction schedules can definitely be a pain especially when changes occur, so with a more lean-based techniques such as Modular units, resourcing those activities, their related risks and other requirements can more determined in alignment Modular-related considerations on-site" (CP 1)<br>"4D BIM has the potential to enhance our scheduling capabilities, and a lean-based model would further emphasize the importance of efficiency in modular unit manufacturing. It would help us streamline our operations and allocate resources more effectively." (MUM 1)<br>"Lean principles align well with the goals of modular unit manufacturing. By incorporating a lean-based 4D BIM model, we can identify and address inefficiencies in our scheduling and resource allocation, ultimately improving our production processes." (MUM 2) |
| Do you think that proposed model would support improved process model for Lean-integrated 4D BIM in Modular Construction? | "Construction scheduling is often complex, and requires a lot of data to provide more informed decision-making, and I suppose that Modular Construction is supposed to minimize many of those complexities, so the proposed 4D BIM process model would support narrowing down many of the risks and uncertainties associated with Modular Construction" (CP 4)<br>"Many of the issues we face resulted from many of the uncertainties, so it would definitely help if 4D BIM Model can support providing early indication of risks such as constructability, operations and logistics, and additional constraints so that construction planners can track schedules more effectively" (CP 3)<br>"In many ways, construction planners do not collaborate with BIM Managers during design development, so involving them would help identifying many of the wastes whether as a result of operations, resources, or even during activity sequencing" (BM 2)<br>"I think that the proposed process would support providing a filtering system especially for critical activities during substructure, superstructure where most issues occur, and I can see that 4D BIM Models would be an excellent enabler to support schedule tracking on the long term" (CP 1)<br>"I think 4D BIM needs a whole new thinking system when applied in Modular Construction, and this is considerably complex, as we still face difficulties in aligning 4D BIM to traditional component based activities, so with your process model, you can remove a lot of schedule wastes, and that can be an effective approach to manage on-site operations and track schedule more effectively" (BM 1)<br>"I think there is a good opportunity to improve the use of 4D BIM with this process model, so if you have a modular unit, your activities in the schedules would be restricted to installation and their relevant health and safety requirements, but would reduce many uncertainties and risks, so I think this will make 4D BIM Models a lot more useful" (CP 2)<br>"Collaboration between construction planners and modular unit manufacturers is crucial. This process model can serve as a bridge to foster better communication and coordination, ultimately enhancing our ability to track schedules more efficiently." (MUM 1)<br>"The proposed process model's emphasis on risk identification aligns well with the challenges we face in modular unit manufacturing. It could provide us with a systematic way to manage uncertainties and improve the overall efficiency of our processes." (MUM 2) |

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
