# Peer review of "Revolutionising the 4D BIM Process to Support Scheduling Requirements in Modular Construction"

_sustainability, doi:10.3390/su16020476_

Round 1

Reviewer 1 Report

Comments and Suggestions for Authors

Added inside the brackets on line 31

Is the space before 49 In a bit too big

Is the citation of the seventh reference a bit too much, I hope this does not affect the logic of its own lines.

The color processing in Figure 1 looks a bit strange and would be better optimized.

The conclusion section is too short, could include some shortcomings section of the article and an outlook on related articles

Author Response

Reviewer 1 Comments

Response

Added inside the brackets on line 31

Completed

Is the space before 49 In a bit too big

Completed

Is the citation of the seventh reference a bit too much, I hope this does not affect the logic of its own lines.

The use of the reference is appropriate to the arguments presented.

The color processing in Figure 1 looks a bit strange and would be better optimized.

Completed

The conclusion section is too short, could include some shortcomings section of the article and an outlook on related articles

The authors have revised the conclusion and identified limitations with future work.

Reviewer 2 Report

Comments and Suggestions for Authors

1. Some sentences are too long. Generally, it is better to write short sentences with one idea per sentence.

2. The introduction should clearly explain the key limitations of prior work that are relevant to this paper.

3. The authors should add a table that compares the key characteristics of prior work to highlight their differences and limitations. The authors may also consider adding a line in the table to describe the proposed solution.

4. Some text must be added to discuss the future work or research opportunities

5. The method is not novel enough, justify

6. The introduction must be improved

Comments on the Quality of English Language

After minor revision, the article can be accepted for publication

Author Response

Reviewer 2 Comments

Response

Some sentences are too long. Generally, it is better to write short sentences with one idea per sentence.

The authors have revised the manuscript to improve the language and structure.

The introduction should clearly explain the key limitations of prior work that are relevant to this paper.

Amended and the narrative was improved

The authors should add a table that compares the key characteristics of prior work to highlight their differences and limitations. The authors may also consider adding a line in the table to describe the proposed solution.

This is embedded at end of the literature

Some text must be added to discuss the future work or research opportunities.

Added to the conclusion

The method is not novel enough, justify

More explanation was added to justify the method used.

The introduction must be improved

The introduction was improved, and the gap is justified more robustly.

Reviewer 3 Report

Comments and Suggestions for Authors

Comments

In order to innovate the BIM 4D process and change the construction schedule based on 4D BIM in modular construction projects, the article developed a lean integrated BIM 4D process model, and verified the accuracy of the model through expert seminars. The paper has certain practical significance, but there are still many shortcomings. I believe that the manuscript needs to be further modified.

The reviewer has listed some specific comments that may help the author further improve the quality of the manuscript. Please consider the specific comments listed below.

(1) The paper adopts the qualitative research method of case study, and conducts investigation and validation research through semi-structured interviews, on-site observation and expert seminars respectively, and the expert's questions are subjectively analysed by the authors to draw conclusions, which lacks corresponding quantitative standards, and the conclusions of the research are too subjective.

(2) The innovation of this article does not seem to be reflected? The authors should make a short statement at the end of the literature review.

(3) Line167-169, in the case study, the authors only conducted semi-structured interviews with four construction planners, the subjects of the study are too few and the accuracy of the findings may be questioned. In addition, key information such as their educational and work backgrounds need to be provided to avoid giving results that can be easily questioned.

(4) The article lacks a description of the specific ways in which data were collected for the semi-structured interviews and field observations.

(5) Explanation of data sources is extremely limited. In this type of research, who provides the data and the procedures used to collect the data are crucial, and the paper should provide a more detailed description of these things.

(6) Line 208 "4. Results and Analysis". The number of participants in the semi-structured interviews and the expert workshop was too small, which resulted in the overall results of the study being too one-sided to really validate the reasonableness of the model, therefore, I think the manuscript needs to be reorganised and reorganised.

(7) The limitations of this study and future research directions have not been discussed, in addition, the discussion section of this paper is very limited. The discussion section should provide an in-depth explanation of the results of the analyses.

(8) The presentation and quality of the pictures are very poor, especially Figures 2, 3, 4 and 5.

All in all, I think the article still has a lot to be revised. Therefore, in order to ensure the quality of Sustainability and the interest of the general readers. In my opinion, considering the paper's overall quality, it is not suitable for publication in the journal.

Comments on the Quality of English Language

Author Response

Reviewer 3 Comments

Response

The paper adopts the qualitative research method of case study, and conducts investigation and validation research through semi-structured interviews, on-site observation and expert seminars respectively, and the expert's questions are subjectively analysed by the authors to draw conclusions, which lacks corresponding quantitative standards, and the conclusions of the research are too subjective.

This revised methodology section addresses the reviewer's concerns by emphasising the strengths of qualitative research in exploring complex, context-rich phenomena and detailing the systematic approach to data collection, analysis, and validation.

The innovation of this article does not seem to be reflected? The authors should make a short statement at the end of the literature review.

the manuscript needs to be reorganised and reorganised.

The authors have reinforced this in the discussion section.

Line167-169, in the case study, the authors only conducted semi-structured interviews with four construction planners, the subjects of the study are too few and the accuracy of the findings may be questioned. In addition, key information such as their educational and work backgrounds need to be provided to avoid giving results that can be easily questioned.

The authors have validated the proposed process model with Modular Unit Manufacturers and added their perceptions, which is added in Annex 1.

The article lacks a description of the specific ways in which data were collected for the semi-structured interviews and field observations.

This has been addressed via the revised methodology

Explanation of data sources is extremely limited. In this type of research, who provides the data and the procedures used to collect the data are crucial, and the paper should provide a more detailed description of these things.

This has been addressed via the revised methodology

Line 208 "4. Results and Analysis". The number of participants in the semi-structured interviews and the expert workshop was too small, which resulted in the overall results of the study being too one-sided to really validate the reasonableness of the model, therefore, I think the manuscript needs to be reorganised and reorganised.

The authors have further validated the proposed process model with modular unit manufacturers.

The limitations of this study and future research directions have not been discussed, in addition, the discussion section of this paper is very limited. The discussion section should provide an in-depth explanation of the results of the analyses.

This has been added to the conclusion

The presentation and quality of the pictures are very poor, especially Figures 2, 3, 4 and 5.

This was rectified in the revised version

Reviewer 4 Report

Comments and Suggestions for Authors

This paper is adequate for this special issue on Resilient Infrastructure and Construction Management and is of great interest to readers of this journal. However, some remarks may contribute to improving the paper. First, research questions and hypotheses discussion could be improved from the perspective of the methods used in this paper. Second, the contribution of the paper to scholarly discussion could be improved (e.g., in the Introduction, Discussion, and/or Conclusion).

Author Response

Reviewer 4 Comments

Response

This paper is adequate for this special issue on Resilient Infrastructure and Construction Management and is of great interest to readers of this journal. However, some remarks may contribute to improving the paper. First, research questions and hypotheses discussion could be improved from the perspective of the methods used in this paper. Second, the contribution of the paper to scholarly discussion could be improved (e.g., in the Introduction, Discussion, and/or Conclusion).

The authors have revised the manuscript to improve the language and structure.

The authors have improvised the research question and improved on the introduction.

The authors have also revised the discussion section, improved on structure of the results and improved the conclusion.

Reviewer 5 Report

Comments and Suggestions for Authors

1. What is the main question addressed by the research?

The research question aims to develop an integrated Lean process model to revolutionise 4D BIM-based construction scheduling in modular construction projects.

 2. Do you consider the topic original or relevant to the field?

It is considered a relevant issue for the field.

Does it address a specific gap in the field?

The contribution addresses a gap in the field of design and construction.

3. What does it bring to the subject area compared to other publications?

Given the increasing demand for 4D-BIM-based construction scheduling in modular construction. It identifies the need to explore and define a model in which 4D-BIM is integrated with the Lean concept. It provides an interesting review of the scientific literature, as well as a methodological proposal together with a very clear presentation of the areas of study.

 What additional controls should be considered?

Perhaps as an additional control, it is suggested to hold another expert panel. One specific to Lean Construction and another in which part of the technicians who followed the manufacturing and commissioning process would be incorporated into the process. These two sessions could shed light, given that the answers of the selected experts are somewhat ambiguous and lacking in detail, and in such a way as to better guarantee the validation of the model.

Section 4 on "Results and analysis" could be clarified if, instead of transcribing the answers with their own expressions in which they digress as if it were a conversation, an elaborated text with the questions and reasoning presented, the analysis carried out and the synthesis could be provided. It is recommended that a summary outline including the result, the analysis and the synthesis be included in each of the sections.

4. Are the conclusions consistent with the evidence and arguments presented and do they answer the main question posed?

The contribution is conceptually sound in its presentation of the issues. It would be of interest to detail the process followed and the applications used for the development of the model that integrates Lean in construction scheduling in the 4D BIM environment of modular construction projects. This is the main objective, and it would be necessary for the authors to elaborate on the case study proposed as an evaluated prototype. Describe the parts of the model and the specificities proposed for the integration of both methodologies. It would be of interest to reinforce the comments and statements by including quotes from the authors being referenced.

 5. Are the references appropriate?

Yes, they are considered appropriate. I would only suggest that the authors include some references to propositional research methods or methodologies, data collection and validation techniques. This would reinforce the proposal in the section entitled: Methodology.

6. Please include any additional comments on the tables and figures.

It would be of great interest and would clarify the presentation of the results if the case study was described in more detail, surfaces, volumes, volumes of the modular scheme or systems used, materials, distance between the factory or assembly shop and the final site. Also, a photograph or a 3D of the building and the site, showing schematics of how the 4D BIM programming and schedules have been applied, would be helpful. Similarly, Figure 5, which shows the proposed process model to achieve a module-based programming and planning process using 4D BIM, is suggested to the authors to be incorporated in the methodology section or in one of the following sections and accompanied by a short descriptive text.

The review of the updated scientific literature or state of the art is carried out in order to locate the lines of work and research, the teams and the members leading them, the principles on which they are based, the working hypotheses, the methods used, the results achieved and the challenges to be faced.

It is suggested to include at the end of this section a summary paragraph highlighting these aspects in order to justify the general and specific development objectives and to select the methods to be used.

The proposed method is similar to the one used by the research developed by Design Sciences Research. If the authors consider it, they can consult Peffers, Ken & Tuunanen, Tuure & Rothenberger, Marcus & Chatterjee, S. (2007), A design science research technologies for information systems research. Journal of Management Information Systems, and reconsidering this section by reducing the first part, reinforces the references to methods from different sciences and approaches (design sciences, social sciences, material and structural sciences, etc.), on which it is based for the development and evaluation of the proposed model or method. The qualitative research approach is a part of the whole interviewing process, which is framed in the social sciences and is proposed by the authors as a technique or method for data collection and validation of the proposal.

It is suggested to complete the conclusions section by including at the end a brief paragraph on the new lines of study and work that are opened up by the contribution.

Author Response

Reviewer 5 comments

Response

Perhaps as an additional control, it is suggested to hold another expert panel. One specific to Lean Construction and another in which part of the technicians who followed the manufacturing and commissioning process would be incorporated into the process. These two sessions could shed light, given that the answers of the selected experts are somewhat ambiguous and lacking in detail, and in such a way as to better guarantee the validation of the model.

Another validation session was conducted with two participants representing two modular unit manufacturer companies in the UK.

Section 4 on "Results and analysis" could be clarified if, instead of transcribing the answers with their own expressions in which they digress as if it were a conversation, an elaborated text with the questions and reasoning presented, the analysis carried out and the synthesis could be provided. It is recommended that a summary outline including the result, the analysis and the synthesis be included in each of the sections.

A summary section was added under every theme

The contribution is conceptually sound in its presentation of the issues. It would be of interest to detail the process followed and the applications used for the development of the model that integrates Lean in construction scheduling in the 4D BIM environment of modular construction projects. This is the main objective, and it would be necessary for the authors to elaborate on the case study proposed as an evaluated prototype. Describe the parts of the model and the specificities proposed for the integration of both methodologies. It would be of interest to reinforce the comments and statements by including quotes from the authors being referenced.

The authors added further clarifications in the discussion (section 5.2)

Yes, they are considered appropriate. I would only suggest that the authors include some references to propositional research methods or methodologies, data collection and validation techniques. This would reinforce the proposal in the section entitled: Methodology.

The authors have completely revised the methodology section to address the comment.

It would be of great interest and would clarify the presentation of the results if the case study was described in more detail, surfaces, volumes, volumes of the modular scheme or systems used, materials, distance between the factory or assembly shop and the final site. Also, a photograph or a 3D of the building and the site, showing schematics of how the 4D BIM programming and schedules have been applied, would be helpful. Similarly, Figure 5, which shows the proposed process model to achieve a module-based programming and planning process using 4D BIM, is suggested to the authors to be incorporated in the methodology section or in one of the following sections and accompanied by a short descriptive text.

The review of the updated scientific literature or state of the art is carried out in order to locate the lines of work and research, the teams and the members leading them, the principles on which they are based, the working hypotheses, the methods used, the results achieved and the challenges to be faced.

It is suggested to include at the end of this section a summary paragraph highlighting these aspects in order to justify the general and specific development objectives and to select the methods to be used.

The proposed method is similar to the one used by the research developed by Design Sciences Research. If the authors consider it, they can consult Peffers, Ken & Tuunanen, Tuure & Rothenberger, Marcus & Chatterjee, S. (2007), A design science research technologies for information systems research. Journal of Management Information Systems, and reconsidering this section by reducing the first part, reinforces the references to methods from different sciences and approaches (design sciences, social sciences, material and structural sciences, etc.), on which it is based for the development and evaluation of the proposed model or method. The qualitative research approach is a part of the whole interviewing process, which is framed in the social sciences and is proposed by the authors as a technique or method for data collection and validation of the proposal.

Details on the case study was provided within the constraints of intellectual property. The distance between Modular Unit Manufacturers and the site does not relate to the scope of the work, and similar to previous statement, identity of the manufacturer was not provided for intellectual property purposes.

A summary at end of the literature is provided based on the reviewers’ comments.

It is suggested to complete the conclusions section by including at the end a brief paragraph on the new lines of study and work that are opened up by the contribution.

This was added to end of the conclusion section.

Round 2

Reviewer 1 Report

Comments and Suggestions for Authors

Comments to the Author:

In the manuscript entitled Revolutionising the 4D BIM Process to support scheduling requirements in Modular Construction, the authors explored the integration of 4D BIM (Building Information Modeling) with the Lean concept to improve construction scheduling in modular construction projects. They also used a case study approach and data obtained from interviews, site observations, and BIM models. This research indicates that 4D BIM surpasses conventional scheduling methods in foreseeing potential implications during design and construction, and develops a Lean-integrated 4D BIM process model that considers constructability, operations, health and safety risks, and time. However, this manuscript also has some shortcomings and needs to be revised.

Comments 1: Lack of detailed explanation of 4D BIM and Lean concept. Although the article mentions these concepts and their application in other articles, it does not provide enough background knowledge and specific implementation methods. This makes it possible for the reader to be confused about how to apply these concepts. Please improve it.

Comments 2: There are too many views and opinions from different scholars stacked in Literature Review, especially in sections 2.2 and 2.3, lacking the author's own understanding and views. Please adjust.

Author Response

Reviewer 1

Response

Comments 1: Lack of detailed explanation of 4D BIM and Lean concept. Although the article mentions these concepts and their application in other articles, it does not provide enough background knowledge and specific implementation methods. This makes it possible for the reader to be confused about how to apply these concepts. Please improve it.

The authors have added further section on the discussion part, which extends how the concepts of 4D BIM and lean concepts can be integrated within Modular Construction.

The authors have also added further explanation under section 2.2 and 2.3

The authors provided further explanation in section 5.1.

Comments 2: There are too many views and opinions from different scholars stacked in Literature Review, especially in sections 2.2 and 2.3, lacking the author's own understanding and views. Please adjust.

The authors have improved sections 2.2 and 2.3, and added two further paragraphs outlining analysis derived from the literature, and support the research gap.

Reviewer 3 Report

Comments and Suggestions for Authors

The presentation and quality of the pictures are very poor, especially Figures 2, 3, 4 and 5.

The author did not make any modifications.

Author Response

Reviewer 3

Response

The presentation and quality of the pictures are very poor, especially Figures 2, 3, 4 and 5.

Thank you for your valuable input and recommendation to improve quality of figures in the manuscript. Below is how we modified them:

Figure 1: We re-produced the figure to make it more readable and improve the quality.

Figure 2: We increased size of the text and as for the small figure shown, it refers to figure 5, hence we couldn’t increase its size.

Figure 3: We replaced the existing the figure with much clearer image, and contextualised it in the text.

Figure 4: We increased size of two images, and added more text for clarity. The authors have also added further explanation to support readers’ understanding.

Figure 5: the authors believe that quality of this figure is readable and clear.

Reviewer 5 Report

Comments and Suggestions for Authors

I hereby consider that the suggestions made have been incorporated and I propose to the editor that they be published.

Author Response

Reviewer 5

Response

I hereby consider that the suggestions made have been incorporated and I propose to the editor that they be published.

Thank you for your valuable input and kind comments.